# Preparation of ESAT6-Fc Fusion Protein and Its Therapeutic Efficacy and Immune Mechanisms in Allergic Asthma Mice via Intranasal Immunization

**DOI:** 10.3390/molecules31010007

**Published:** 2025-12-19

**Authors:** Jing Wang, Maiyan Hai, Yuxin Yang, Tiansong Wang, Wei Zhang, Rui Ma, Miao Sun, Yanyan Qin, Yuan Yang, Zihan Dong, Maosheng Yang, Qiaofeng Wan

**Affiliations:** 1Department of Pathogenic Biology and Immunology, College of Basic Medical Science, Ningxia Medical University, Yinchuan 750004, China; 17835684863@163.com (J.W.); hmyhmy0821@126.com (M.H.); yangyuxin19971002@126.com (Y.Y.); wencheng1998211@163.com (T.W.); zhangwei@nxmu.edu.cn (W.Z.); mr_312@126.com (R.M.); nxsunm@126.com (M.S.); 18295580390@163.com (Y.Q.); yy294711083@163.com (Y.Y.); dongzihan2004@163.com (Z.D.); 2Key Laboratory of Prevention and Control of Common Infectious Diseases in Ningxia, Yinchuan 750004, China; 3Department of Natural Resource Management and Environmental Geography, College of Geographic Sciences and Planning, Ningxia University, Yinchuan 750021, China

**Keywords:** Allergic Asthma (AA), Mycobacterium tuberculosis (Mt.b), ESAT6-Fc, intranasal immunization, integrated transcriptomic and proteomic analysis

## Abstract

The respiratory mucosal system plays a critical role in the pathogenesis of allergic asthma (AA). Currently, therapeutic Fc fusion proteins are as a promising strategy for mucosal vaccine delivery systems. In this work, a plasmid encoding the Mycobacterium tuberculosis ESAT6-Fc fusion protein was successfully constructed, and high-purity ESAT6-Fc fusion protein was subsequently obtained. Administered via intranasal immunization in OVA-induced allergic asthma model mice, ESAT6-Fc fusion protein significantly alleviated airway inflammation and mucus production, and reduced the proportions of Th2 cells, Th17 cells, and eosinophils, while increasing the proportions of Th1 cells with no histopathological changes to major organs. To elucidate the underlying immune regulatory mechanisms of ESAT6, integrated transcriptomic and proteomic analyses were performed, revealing Th1/Th2 cell differentiation and Th17 cell differentiation as the two most significantly enriched pathways at both the gene and protein levels. CD3e (*CD3E*) and CD3g (*CD3G*), two essential subunits of the TCR–CD3 complex, were identified as core target factors. The validations from the ESAT6-Fc-treated AA lung tissues, as well as co-cultured TH0 cells from C57BL/6J mice and CD2.4 dendritic cells exposed to the ESAT6-Fc protein, were consistent with the aforementioned findings. ESAT6-Fc exhibits a safe profile with favorable efficacy against OVA-induced AA via intranasal immunization, and ESAT6 ameliorates AA by regulating the differentiation of Th0 cells into Th1 cells, which were closely associated with the down-regulation of CD3e and CD3g expression, presumably leading to the impairment of TCR–CD3 complex assembly. ESAT6-Fc fusion protein demonstrates promise as a potential safe intranasal immunotherapy agent for the treatment of AA.

## 1. Introduction

Allergic Asthma (AA) is characterized by elevated levels of IgE; type 2 cytokines, including interleukin-4 (IL-4) and IL-13; Th17 cytokines, such as interleukin-17A-F, IL-21, and IL-22; mucus hypersecretion; eosinophilia; and airway hyperresponsiveness (AHR) [1,2]. Dendritic cells (DCs) are the most potent antigen-presenting cells of the immune system and act as a bridge between innate and adaptive immunity [3]. Upon stimulation by an allergen, dendritic cells (DCs) promote the differentiation of naïve T cells into T helper 2 (Th2) and T helper 17 (Th17) cells [4]. Moreover, Th2 cytokines, such as interleukin-4 (IL-4) and interleukin-13 (IL-13), contribute to the recruitment and activation of eosinophils in the airways [5]. Th17 cytokines, including interleukin (IL)-17A to IL-17F, IL-21, and IL-22, induce mucous cell metaplasia and exert pleiotropic effects on airway smooth muscle, contributing to airway narrowing [2,6]. Meanwhile, Th2 cells promote IgE production through their interaction with B cells. The produced IgE binds to high-affinity type I IgE receptors (FcεRI) on mast cells and basophils, thereby sensitizing these cells to respond upon subsequent exposure to the same allergen [7]. This process effectively leads to the release of a wide range of inflammatory mediators, such as histamine (His) and leukotrienes (LTB), thereby contributing to the development of AHR [8,9].

In recent decades, the incidence of allergic diseases in industrialized countries has been steadily increasing [10], and effective treatment may involve the regulation of allergic inflammatory responses [11]. The “Hygiene Hypothesis” has evolved into the “Microbiota Hypothesis,” which posits that early microbial exposures may influence the risk of developing allergic diseases and asthma later in life [12]. Epidemiological studies have indicated a negative correlation between the prevalence and severity of atopic asthma and *Bacillus* Calmette–Guérin (BCG) vaccination [13,14]. Nevertheless, the mechanism underlying the inhibitory effects of *Mycobacterium tuberculosis* (*MTB*) components on allergic diseases remains poorly understood. ESAT6 (early-secreted antigenic target of 6 kDa), an immunoregulatory protein and potent T cell antigen, is secreted through the ESX-1 secretion system (type VII) of *MTB* [15], which is one of the major factors involved in modulating host immune responses [16] and altering host immunity through regulating the development or activation of various immune cells [17]. A previous study has reported the effect of ESAT6-Like Protein ESXV (encoded by *Rv3619c*) on AA in mice [18]; however, the immune regulatory mechanisms through which ESAT6 may alleviate the progression of AA remain unclear and warrant further investigation.

Specifically, IgGs are characterized by their high circulating concentrations, extended serum half-lives, and the unique capacity to be transferred from mother to offspring through interactions with the neonatal Fc receptor (FcRn) [19]. This receptor specifically binds to the Fc region of IgG molecules, thereby facilitating the transcellular transport of IgG across polarized cell layers, such as those found in the endothelium and epithelium [20]. Based on the aforementioned principle, a therapeutic Fc fusion protein, which comprises the Fc region of IgG and a target protein, has been developed for the treatment of various diseases, including AIDS [21], hemophilia [22], and systemic lupus erythematosus [23].

Given that intranasal administration is preferred due to its direct access to target sites, minimal invasiveness, and rapid onset of action, this study aimed to evaluate the therapeutic efficacy of the ESAT6-Fc fusion protein when administered via intranasal mucosa in OVA-induced AA model mice, and an integrated analysis of transcriptomics and proteomics was subsequently conducted to investigate the potential immune regulatory mechanisms.

## 2. Results

### 2.1. Effects of ESAT6-Fc Fusion Protein on Allergic Asthma Model Mice

The effects of the ESAT6-Fc fusion protein on OVA-induced AA were assessed. Firstly, the eukaryotic expression plasmid encoding the ESAT6-Fc fusion protein was successfully constructed, and the ESAT6-Fc fusion protein was expressed in CHO cells, with a purity exceeding 95% (Figure 1A). Subsequently, in order to evaluate the therapeutic potential of the ESAT6-Fc fusion protein in alleviating AA, the OVA model was established and treatment interventions were administered (Figure 1B). In the OVA group, the percentages of eosinophils (Eos), Th2 cells, and Th17 cells were significantly elevated, whereas the percentage of Th1 cells was significantly reduced compared to that in the Normal group (Figure 1C; *p* < 0.001 for Eos, *p* < 0.0001 for Th1, *p* < 0.001 for Th2 and *p* < 0.0001 for Th17). These findings suggest that 1% OVA effectively activated immune signaling pathways to mimic AA. There was no significant difference in the percentage of cells between the OVA group and the CpG group (*p* > 0.05). In contrast, both the ESAT6-Fc+CpG group and the DEX group exhibited a significant reduction in the percentage of Eos (both *p* < 0.05), Th2 (both *p* < 0.05), and Th17 cells (both *p* < 0.001), along with a significant increase in the proportion of Th1 cells (*p* < 0.0001 and *p* < 0.001, respectively). No significant differences were observed between the ESAT6-Fc+CpG and DEX groups (*p* > 0.05). These findings suggest that the immunomodulatory role of ESAT6-Fc + CpG is attributed to the ESAT6-Fc, which may suppress the activation of certain immune signaling.

To assess whether the ESAT6 protein modulates the molecular expression involved in AA, the levels of His, LTB4, IgE, IL-4, IL-13, IL-17A, and IFN-γ in BALF were tested using ELISA. The results demonstrated that the levels of His, LTB4, IgE, IL-4, IL-13, and IL-17A were significantly elevated, while IFN-γ was decreased in the OVA group mice compared to the Normal group mice (Figure 1D; *p* < 0.0001 for all comparisons). There was no significant difference in levels of these molecules between the OVA group and the CpG group (*p* > 0.05), with the exception of IgE (*p* < 0.05). In contrast, both the ESAT6-Fc+CpG group and the DEX group showed significantly reduced levels of His (*p* < 0.001, *p* < 0.01), LTB4 (both *p* < 0.01), IgE (both *p* < 0.001), IL-4 (both *p* < 0.01), IL-13 (*p* < 0.0001, *p* < 0.001) and IL-17A (both *p* < 0.001), and elevated levels of IFN-γ (both *p* < 0.001). No significant differences in the levels of these molecules were observed between the ESAT6-Fc+CpG and DEX groups (*p* > 0.05). These findings indicate that intranasal administration of ESAT6-Fc can achieve therapeutic levels comparable to those of DEX.

In addition, the histopathological analysis indicated remarkable leukocyte infiltration in the peribronchial region of the lung tissue of mice in the OVA group. Meanwhile, the infiltration level of leukocytes was markedly reduced in the ESAT6-Fc+CpG group mice (Figure 1E). Concurrently, histological examination revealed a significant increase in periodic Acid-Schiff (PAS)-positive cells in the OVA group, indicating enhanced mucus secretion. In contrast, this response was markedly attenuated in the ESAT6-Fc+CpG treatment group, suggesting that ESAT6-Fc significantly suppressed mucus production (Figure 1F). Furthermore, HE staining and measuring of AST and LDH revealed no histopathological changes in the heart, liver, spleen, or kidney samples of mice in the CpG and ESAT6-Fc+CpG groups compared to the Normal group (Appendix A), thus highlighting the safety profile of ESAT6-Fc.

This study demonstrated that ESAT6-Fc exhibits a safe profile with favorable efficacy against OVA-induced AA via intranasal immunization. To further elucidate the immunomodulatory mechanisms underlying ESAT6-Fc action, integrated transcriptomic and proteomic analyses were conducted on lung tissues from three experimental groups: Normal, OVA, and ESAT6-Fc (the original ESAT6-Fc+CpG).

### 2.2. Differentially Expressed Genes (DEGs) and Proteins (DEPs) Between the Normal and OVA Groups

To validate the successful establishment of the AA model, an integrated transcriptomic and proteomic profiling approach was employed. Comparative analyses between the Normal and OVA groups, based on RNA sequencing and tandem mass tag (TMT)-based quantitative proteomics, revealed 614 differentially expressed genes (DEGs), of which 530 were significantly upregulated and 84 were downregulated. Concurrently, 149 differentially expressed proteins (DEPs) were identified, including 117 upregulated and 32 downregulated proteins (Figure 2A,B). Notably, the DEGs were primarily enriched in the biological processes related to cytokine–cytokine receptor interactions, cell cycle regulation, cytotoxicity mediated by natural killer cells, and T cell receptor signaling pathways (Figure 2C), whilst DEPs were primarily enriched in the signaling pathways involved in asthma, Th1 and Th2 cell differentiation, as well as Th17 cell differentiation (Figure 2D). The activation of these key signaling pathways in mediating allergic inflammation [24,25] provides strong evidence for the successful establishment of the AA model.

### 2.3. DEGs and DEPs After Treatment with the ESAT6-Fc Fusion Protein

To elucidate the signaling pathways modulated by the ESAT6 protein in the AA model, we conducted a comparative KEGG pathway analysis using transcriptomic and proteomic data derived from the OVA and ESAT6-Fc groups. Following treatment with the ESAT6-Fc fusion protein, 520 differentially expressed genes (DEGs) and 147 differentially expressed proteins (DEPs) were identified. Among the DEGs, 52 were upregulated and 468 were downregulated (Figure 3A), whereas among the DEPs, 42 were upregulated and 105 were downregulated (Figure 3B). Integrated multi-omics enrichment analysis revealed significant enrichment of pathways associated with Th1/Th2 cell differentiation and Th17 cell differentiation at both transcriptional and translational levels. The expression patterns and counts of DEGs and DEPs within these significantly enriched pathways are presented in Figure 3E,F. In the Th1/Th2 cell differentiation pathway, 17 genes and 5 proteins were downregulated, compared to 1 gene and 1 protein that were upregulated. Similarly, in the Th17 cell differentiation pathway, 15 genes and 4 proteins were downregulated, while 1 gene and 1 protein were upregulated.

### 2.4. DEGs and DEPs Regulated by the ESAT6 Protein

To investigate the immunomodulatory mechanisms underlying the effect of the ESAT-6 protein on AA, key regulatory factors at the gene and protein levels were identified as being involved in two significantly enriched pathways, Th1 and Th2 cell differentiation and Th17 cell differentiation. The DEGs and DEPs that were co-downregulated by the ESAT6 protein in both pathways include *CD3G* (CD3g) and *CD3E* (CD3e), whilst no DEGs and DEPs were co-upregulated. *CD3G* (CD3g) and *CD3E* (CD3e) are two distinct heteropolymers that constitute the CD3 complex [26]. The clonotypic αβ TCR heterodimer, noncovalently associated with invariant CD3 dimers (CD3 ϵ γ, CD3 ϵ δ, and CD3ζζ), forms a functional complex that enables T cells to recognize specific peptide–MHC (pMHC) ligands through the TCR, thereby initiating intracellular activation signaling via the CD3 subunits [27]. It was evident that ESAT6 mitigates the inflammatory response of AA by downregulating the gene and protein expression of *CD3E* (CD3e) and *CD3G* (CD3g), and further suppressing the assembly of TCR–CD3 complexes and their signal transduction.

Taken together, CD3e (*CD3E*) and CD3g (*CD3G*) were identified as core targets regulated by ESAT6 via Th1/Th2 cell differentiation and Th17 cell differentiation pathways, and their molecular interaction network is displayed in Figure 4A,B.

### 2.5. Validation Experiments 

In the animal studies, Western Blot analysis demonstrated that ESAT6-Fc significantly downregulated the levels of CD3e and CD3g in the lung tissues of AA mice (both *p* < 0.01) (Figure 5A); and immunofluorescence analysis revealed that the ESAT6-Fc clearly reduced the number of CD3+T cells in the lung tissues of AA mice (Figure 5B).

In the vitro cell experiments, ESAT6-Fc promoted the differentiation of Th0 cells into Th1 cells (*p* < 0.01), while simultaneously inhibiting their differentiation into Th2 and Th17 cell subsets (both *p* < 0.001) (Figure 5C); Western Blot analysis demonstrated that ESAT6-Fc downregulated the levels of CD3e and CD3g in the CD4+T cells (both *p* < 0.01) (Figure 5D); ELISA results demonstrated that ESAT6-Fc elevated the levels of IFN-γ and IL-12 (*p* < 0.001, *p* < 0.0001, respectively) while reducing the level of IL-17A (*p* < 0.001) in the co-culture system (Figure 5E). Cellular experiments demonstrated that ESAT6 facilitated the differentiation of Th0 cells into Th1 cells, which was associated with the down-regulation of CD3e and CD3g expression.

## 3. Discussion

Asthma affects 3% of the global population, leading to over 0.25 million deaths annually [28]. As the main phenotype of asthma, allergic asthma (AA) imposes heavy economic and productivity burdens [29]. However, the conventional treatment modalities used for AA are associated with side effects [30,31], thus emphasizing the need for alternative therapeutic agents.

CD3-positive T lymphocytes are elevated in patients with allergic diseases [32], and CD4 T cells play critical roles in mediating adaptive immunity-involved AA responses [33]. During TCR activation, naive CD4+ T cells (Th0) differentiate into one of several lineages of T helper (Th) cells, including Th1, Th2, and Th17 cells, among others [34,35]. Th cell differentiation involves a two-phase process. The first phase is the T cell receptor (TCR)-driven induction phase, during which key transcription factors are induced or activated; the second is the cytokine-driven polarization phase, in which the expression of these key factors is amplified and the differentiation process is completed [36]. Each subset is governed by a master regulatory transcription factor and a specific signal transducer and activator of transcription (STAT). Th2 cells are associated with GATA-3 and STAT5; Th1 cells with T-bet and STAT4; and Th17 cells with retinoid orphan receptor γt (RORγt) and STAT3 [37]. Both Th2 and Th17 cells have been implicated in the pathogenesis of AA [38,39]. Currently, T cell-mediated immunotherapy represents a promising therapeutic strategy for various allergic diseases, including AA [40,41]. Given that the CD3 complex is a key molecule involved in modulating T cell function, monoclonal antibody-based therapies targeting CD3 are being investigated and applied in the treatment of AA [42].

The respiratory mucosal system plays a critical role in the pathogenesis of AA by serving as a primary defensive barrier. The epithelial lining protects the host by preventing the entry of airborne environmental particles [43]. In recent years, respiratory mucosal immunotherapy has emerged as a pivotal component of the treatment of asthma and other diseases [44,45]. In this study, the ESAT6-Fc fusion protein was administered intranasally and significantly alleviated airway inflammation and mucus secretion, reduced the proportions of eosinophils, Th2 cells, and Th17 cells, while increasing the proportion of Th1 cells in the lungs. Additionally, the ESAT6-Fc fusion protein significantly reduced the levels of histamine, leukotriene B4, IgE, IL-4, IL-13, and IL-17A, and elevated the level of IFN-γ in the bronchoalveolar lavage fluid of asthmatic mice. Inspired by these findings, we aimed to elucidate the immunoregulatory mechanisms underlying ESAT6. To this end, we conducted a systematic analysis of differentially expressed genes (DEGs) and proteins (DEPs) derived from omics datasets to identify significantly enriched pathways and key molecular targets. A joint two-omics KEGG pathway enrichment analysis uncovered that Th1/Th2 cell differentiation, and Th17 cell differentiation pathways were significantly enriched at both the gene and protein levels. The DEGs and DEPs co-downregulated by the ESAT6 protein in both pathways include *CD3E* (CD3e) and *CD3G* (CD3g).

CD3e or CD3g is the primary transmembrane protein on T cells and has four isoforms, namely CD3d, CD3e, CD3g, and CD3ζ. Its CD3d/CD3e and CD3g/CD3e heterodimers, CD3ζ/CD3ζ homodimer, and the α and β strands of the T cell receptor (TCR) form a TCR–CD3 complex [46]. The integrity of the TCR–CD3 complex is crucial for T cell development and regulation. A defect in either the CD3e or CD3g chain gene can lead to impaired T cell signal transduction [47].

Following FcRn-mediated transport across the respiratory epithelium in AA mice, the ESAT6-Fc fusion protein is internalized through pinocytosis or receptor-mediated endocytosis by submucosal phagocytes, including dendritic cells (DCs), neutrophils, and macrophages. Subsequently, the ESAT6 antigen is processed and presented by DCs, leading to the initiation of a Th1-polarized adaptive immune response [48,49]. Therefore, the proportions of Th2 cells, Th17 cells, and eosinophils in the lungs, as well as the levels of His, LTB4, IgE, IL-4, IL-13, and IL-17A in the BALF, are significantly reduced.

Overall, ESAT6 exerts an anti-AA effect by suppressing the activation of OVA-induced signaling pathways involved in Th1/Th2 and Th17 cell differentiation. The promotion of TH0 cell differentiation into Th1 cells, along with the inhibition of TH0 differentiation into Th2 and Th17 cells mediated by ESAT6, is associated with the downregulation of *CD3E* (CD3e) and *CD3G* (CD3g) at both gene and protein expression levels in TH0 cells through antigen-presenting cells (APCs), which may presumably impair the assembly and functional integrity of the TCR–CD3 complex. The validation results from ESAT6-Fc-treated AA lung tissue, as well as from co-cultures of TH0 cells derived from C57BL/6J mice and CD2.4 cells exposed to the ESAT6-Fc protein, were consistent with the aforementioned findings. The mechanism by which ESAT6 regulates the differentiation of Th0 cells into Th1 cells and simultaneously downregulates the expression of CD3g and CD3e remains to be elucidated.

In this study, a preliminary investigation was conducted into assessment of the therapeutic efficacy of ESAT6-Fc in OVA-induced AA and its potential immunological mechanisms. Furthermore, it was observed that both ESAT6 and ESXV elicited Th1-type immune responses. In future studies, to comprehensively evaluate the effects of ESAT6-Fc on AA, three specific strategies may be employed in animal experiments using the AA model: (1) ESAT6-Fc could be administered to AA model mice induced by other allergens, such as pollen and dust mites; (2) ESAT6-Fc could be tested in other animal species, including guinea pigs and rabbits; (3) The therapeutic efficacy of ESAT6-Fc and ESXV-Fc could be evaluated.

Nevertheless, the limitations of this study should be acknowledged. There is a lack of longitudinal studies exploring immune responses and asthma progression over time with ESAT6-Fc treatment. In the future, longitudinal studies should be conducted to observe the long-term effects of ESAT6-Fc treatment and assess some delayed responses or side effects that may arise over time.

## 4. Materials and Methods

### 4.1. Construction of pcDNA3.1(+)-Rv3875-Fc and Preparation of ESAT6-Fc Fusion Protein

To generate the ESAT6-Fc fusion protein, an expression plasmid pcDNA3.1(+)-Rv3875-Fc was constructed. A synthetic 1116 bp DNA fragment encoding the signal peptide of herpes simplex virus type 2 glycoprotein D (HSV2-gD), the Mycobacterium tuberculosis ESAT6 antigen, and the Fc region of mouse IgG2a was commercially synthesized by Sangon Biotech Co., Ltd. (Shanghai, China). This fragment was cloned into the HindIII and XhoI restriction sites within the multiple cloning site of the pcDNA3.1(+) vector. The resulting recombinant plasmid, designated pcDNA3.1(+)-Rv3875-Fc, was verified through restriction enzyme digestion using HindIII and XhoI (New England Biolabs, Hertfordshire, UK) and by DNA sequencing analysis conducted by Beijing Ruibo Xingke Biotechnology Co., Ltd. (Beijing, China). Afterwards, the ESAT6-Fc fusion protein was expressed in CHO cells [50] and purified using Protein G beads provided by Nanjing Mingyan Biotechnology Co., Ltd. (Nanjing, China). Finally, the presence of the ESAT6-Fc fusion protein was confirmed via Western blot using a rabbit polyclonal antibody specific to ESAT6 (ab45073, Abcam, Cambridge, UK), and the protein purity was assessed by Coomassie staining (Figure 1A).

### 4.2. Animal Grouping, Allergic Asthma Mouse Model and Drug Treatment

A total of 30 female 7-week-old C57BL/6J (weight, 18 ± 1.0 g) mice were purchased from the Experimental Animal Center of Ningxia Medical University (certificate no. SCXK, 2020-0001), and were approved by the Ethics Committee of Ningxia Medical University (approval no. 2025-N094). The mice were raised in a standard pathogen-free environment with a temperature of 22 ± 2 °C, humidity of 55 ± 5%, 12 h light/dark cycles and free access to food and water. The mice were randomly assigned into five experimental groups: (1) Normal group; (2) OVA group (Ovalbumin, allergen); (3) CpG group (CpG-ODN immune adjuvant); (4) ESAT6-Fc+CpG group; and (5) DEX group (Dexamethasone, positive control group). Briefly, with the exception of the Normal group, mice in all other groups were intraperitoneally injected with 1% ovalbumin (OVA, Solarbio, Beijing, China) at seven-day intervals, for a total of three administrations over a 14-day period to induce sensitization [51]. Subsequently, on day 15, mice in the Normal and OVA groups received 20 μL of phosphate-buffered saline (PBS) via intranasal administration; mice in the CpG group were intranasally administered 20 μL of CpG solution containing 10 μg of CpG adjuvant (ODN1826, Invivogen, Toulouse, France); mice in the ESAT6-Fc+CpG group were intranasally administered 20 μL of a mixture consisting of 10 μg of ESAT6-Fc fusion protein and 10 μg of CpG adjuvant; mice in the DEX group were intranasally administered 20 μL of DEX solution containing 20 μg of dexamethasone (H41020035, Sinopharm Ronshyn Pharmaceutical Co., Ltd., Nanyang, China). On the 22nd day, a second administration was carried out [52]. Finally, the mice in the CpG, ESAT6-Fc+CpG, and DEX groups were nebulized with 1% OVA for 20 min once daily from day 29 to day 33 [51]. The treatment timelines are presented in Figure 1B. Following sensitization, the mice were euthanized, and bronchoalveolar lavage fluid (BALF), serum, and tissue samples from the lung, heart, liver, spleen, and kidney were collected for subsequent experimental analyses.

### 4.3. Flow Cytometry Analysis

In the animal studies, right lung lobes were sliced into fragments and digested in RPMI-1640 medium containing 1 mg/mL of collagenase I (D8140, Solarbio, China) and 10 μg/mL of deoxyribonucnaseI (D8071, Solarbio, China) for 60 min. Single-cell suspensions derived from animal lung tissue and from CD4+T cells co-cultured with CD2.4 and ESAT-6-Fc were centrifuged, the supernatant was subsequently removed, and the cells were resuspended in a staining buffer (E-CK-A107, Elabsence, Beijing, China) and labeled with Efluor450-CD45, APC-CD11b, and PE-CD170 (Thermo Fisher, Waltham, MA, USA) for eosinophils; FITC-CD4 and APC-IFN gamma (Thermo Fish, USA) for Th1; FITC-CD4 and Percp-efluor710-IL-4 (Thermo Fisher, USA) for Th2; and PE-IL-17A (Thermo Fisher, USA) for Th17. The percentage of eosinophils, Th1, Th2, and Th17 cells was analyzed using a BD FACSCelesta Flow Cytometer (Becton, Dickinson and Company, Franklin Lakes, NJ, USA).

### 4.4. Enzyme-Linked Immunosorbent Assay (ELISA)

A total of 10 μL of BALF sample was added to each well and subsequently diluted to a final volume of 50 μL for the His, LTB4, IgE, IL-4, IL-13, IL-17A, and IFN-γ ELISA kits (LCSJZF20879, ED20158, ED20508, ED20186, ED20167, ED20170, ED20140, respectively; Hubei Hejun Biotechnology Co., Ltd., Wuhan, China). A total of 50 μL of tissue homogenate from heart, liver, spleen, and kidney tissues of the Normal, CpG, and ESAT-6-Fc+CpG groups was added to each well in the AST and LDH ELISA kits (ED21534, ED28590, respectively; Hubei Hejun Biotechnology Co., Ltd., China). Additionally, 50 μL of CD4+T cell co-cultured supernatant with CD2.4 and ESAT-6-Fc was added to each well in the IL-12, IFN-γ, and IL-17A ELISA kits (ED22503, ED20140, ED20170, respectively; Hubei Hejun Biotechnology Co., Ltd., China). The concentrations of His, LTB4, IgE, IL-4, IL-13, IL-17A, AST, LDH, IL-12, and IFN-γ were determined in accordance with the manufacturer’s instructions.

### 4.5. Histological Analysis

The left lungs were fixed in 4% paraformaldehyde (PFA) for 24 h at room temperature, embedded in paraffin, and subsequently sectioned into 5 μm slices. The paraffin was removed, and the tissue sections were rehydrated before being stained with hematoxylin and eosin (H & E). The degree of inflammatory infiltration in the H&E-stained lung sections was assessed using a semi-quantitative scoring system: “0” indicates no inflammatory cell infiltration; “1” indicates few inflammatory cells; “2” represents a single layer of inflammatory cells surrounding the lesion; “3” denotes 2–4 layers of infiltrating inflammatory cells; and “4” signifies more than four layers of inflammatory cell infiltration [53]. Goblet cell hyperplasia in the airway epithelium with Periodic Acid-Schiff (PAS) stain was assessed using a five-point scoring system: “0” indicates no goblet cells; “1”, less than 25% of the epithelium; “2”, 25–50% of the epithelium; “3”, 51–75% of the epithelium; and “4”, more than 76% of the epithelium [54].

### 4.6. cDNA Library Construction and Transcriptomic Analysis

Total RNA from the lung tissue of each mouse was extracted with the TRIzol Reagent (Invitrogen, Carlsbad, CA, USA). RNA integrity, purity, and concentration were assessed using an Agilent 2100 Bioanalyzer (Agilent Technologies, Santa Clara, CA, USA) and a NanoDrop 2000 spectrophotometer (Thermo Scientific, USA). Samples meeting the required quality criteria were used to construct cDNA libraries with the VAHTS Universal V6 RNA-seq Library Prep Kit, which were subsequently sequenced by Shanghai OE Biotechnology Co., Ltd. (Shanghai, China). Differentially expressed genes (DEGs) were identified using a fold change threshold of >2 and a significance level of *p* < 0.05. Functional pathway analysis of DEGs was performed using the Kyoto Encyclopedia of Genes and Genomes (KEGG) database. Raw Illumina sequences were uploaded to the Gene Expression Omnibus (GEO) database (accession number: GSE274427).

### 4.7. Tandem Mass Tag (TMT)-Labeled Proteomics and Protein Quantification

Protein expression in animal lung tissues was quantified using TMT-based proteomics (Shanghai OE Biotechnology Co., Ltd., Shanghai, China). After protein digestion, peptides were desalted (Strata X C18 SPE column, Phenomenex, Torrance, CA, USA) and TMT-labeled. The labeled peptides were fractionated by HPLC (Agilent 300 Extend C18 column), dissolved in acetonitrile, and analyzed via Q Exactive HF mass spectrometry (Thermo Fisher Scientific, Bremen, Germany). Raw data were processed with Proteome Discoverer 2.4 (Thermo Fisher Scientific) to identify differentially expressed proteins (DEPs), defined as |FC| > 1.5 and *p* < 0.05. DEPs were functionally analyzed using the Kyoto Encyclopedia of Genes and Genomes (KEGG), and the dataset was deposited to Proteome Xchange (accession number: PXD062711).

### 4.8. Cells and Cell Treatment

The mouse dendritic cell line DC2.4 (AY0006) was purchased from Shanghai Huiying Biological Technology Co., Ltd. (Shanghai, China). TH0 cells were isolated from the spleens of C57BL/6 mice using the EasySep™ Mouse CD4+T Cell Isolation Kit (19852) provided by STEMCELL Technologies (Vancouver, BC, Canada). The experiment was divided into two groups—the Th0+CD2.4 group and the Th0+CD2.4+ESAT6-Fc group—with three replicate dishes per group. Briefly, CD2.4 cells were seeded into 6 cm diameter plates at a density of 1 × 10^6^ cells per well. Following cell adhesion, TH0 cells were added to the CD2.4 cell culture dishes at the same density. Simultaneously, the ESAT6-Fc fusion protein was introduced into the co-culturing system at a working concentration of 500 ng/mL, and the cells were co-cultured for 24 h. Finally, CD4+T cells were harvested and processed for flow cytometry analysis and Western Blot. Cell culture supernatants were collected to assess cytokine levels using ELISA.

### 4.9. Western Blot Analysis

Proteins were extracted from animal samples and CD4+T cells using a cell lysis buffer supplemented with protease inhibitor (A8260-5, Amresco, Rancho Cucamonga, CA, USA). Protein concentrations were determined using a BCA protein assay kit (P0010S, Beyotime, Shanghai, China). Subsequently, the proteins were separated by 10% SDS-PAGE and transferred onto PVDF membranes. The membranes were blocked with 5% non-fat milk and incubated overnight at 4 °C with primary antibodies specific to anti-CD3G (ET7107-55, Diagbio, Wuhan, China), anti-CD3E (db12819, Diagbio, China), and anti-β-Actin (AC026, ABclonal, Wuhan, China). Following TBST washing, the membranes were subsequently incubated with an HRP-conjugated secondary antibody (SA00001-2, Proteintech, Rosemont, IL, USA) for 1 h at 37 °C. Signal detection was then carried out using the New Super ECL Assay Kit (KEP1128, KeyGentec, Nanjing, China) in accordance with the manufacturer’s instructions.

### 4.10. Fluorescence Analysis

Deparaffinized tissue sections were subjected to heat-mediated antigen retrieval through incubation at 60 °C for 3 h. Following blocking with 3% bovine serum albumin (BSA) for 30 min, the sections were incubated with the primary antibody at 4 °C overnight in a humidified chamber. After washing with phosphate-buffered saline (PBS), the sections were incubated with a fluorescent secondary antibody (SA00003-2, Proteintech, USA) for 1 h at room temperature. Nuclei were counterstained with 4′,6-diamidino-2-phenylindole (DAPI; S2110, Solarbio, China) during mounting, and fluorescence imaging was performed using an Olympus BX51 fluorescence microscope (Hachioji-shi, Japan).

### 4.11. Statistical Analysis

All data are presented as mean ± standard deviation (SD). Statistical analyses were conducted using GraphPad Prism 6 (GraphPad Software, Inc., San Diego, CA, USA) and the OE Cloud platform (https://cloud.oebiotech.com (accessed on 18 June 2025)). For comparisons among the three experimental groups, one-way analysis of variance (ANOVA) was employed, followed by Dunnett’s post hoc test for pairwise inter-group comparisons. A *p*-value < 0.05 was considered statistically significant.

## 5. Conclusions

In this study, pcDNA3.1(+)-Rv3875-Fc was successfully constructed, and a high-purity ESAT6-Fc fusion protein was obtained. This study demonstrated that ESAT6-Fc exhibits a safe profile with favorable efficacy against OVA-induced AA via intranasal immunization, and ESAT6 ameliorates AA by regulating the differentiation of Th0 cells into Th1 cells, which were closely associated with the downregulation of CD3e and CD3g expressions, presumably leading to impairment of TCR–CD3 complex assembly. ESAT6-Fc fusion protein demonstrates promise as a potential safe intranasal immunotherapy agent for the treatment of AA.

## Figures and Tables

**Figure 1 molecules-31-00007-f001:**
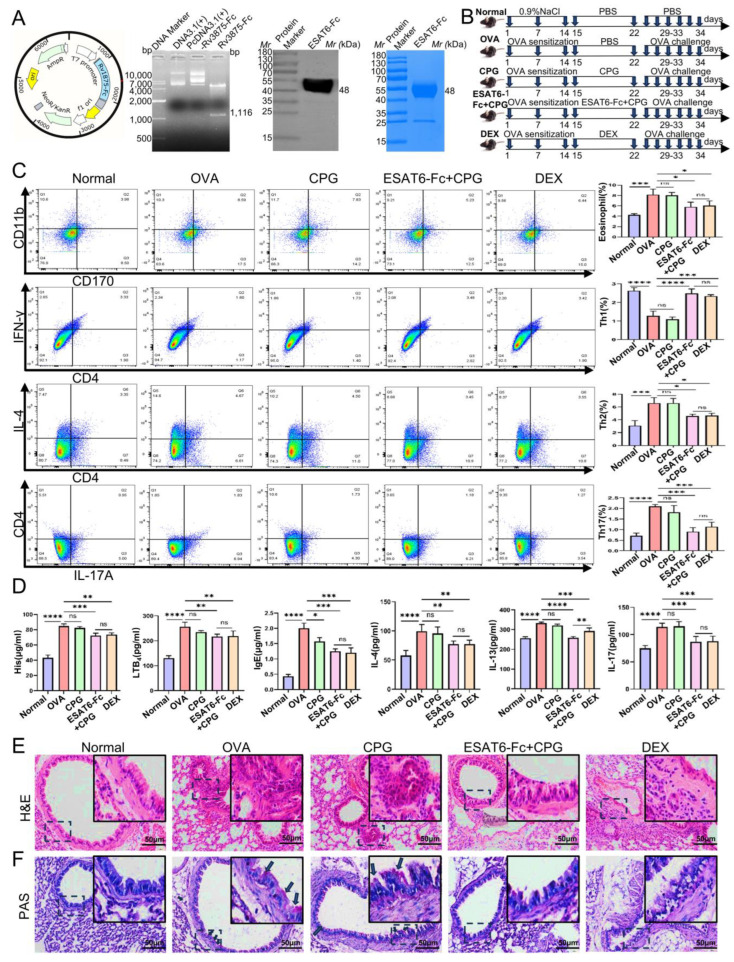
Effects of ESAT6-Fc fusion protein administered via intranasal immunization on OVA-induced AA model mice. (**A**) Construction and enzyme digestion identification of plasmid pcDNA3.1(+)Rv1875-Fc, and separation-purification and purity assessment of ESAT6-Fc fusion protein. (**B**) The treatment timelines for five groups. (**C**) Percentage of eosinophils, Th1 cells, Th2 cells and Th17 cells (*n* = 3). (**D**) Levels of His, LTB4, IL-4, IL-13, IL-17A, IgE and IFN-γ in BALF (*n* = 5). (**E**) Assessment of ESAT6-Fc on airway inflammation in peribronchial region. (**F**) Goblet cell which indicated by the arrow hyperplasia following ESAT6-Fc treatment. Data are expressed as the mean ± standard deviation (SD). * *p* < 0.05, ** *p* < 0.01, *** *p* < 0.001 and **** *p* < 0.0001.

**Figure 2 molecules-31-00007-f002:**
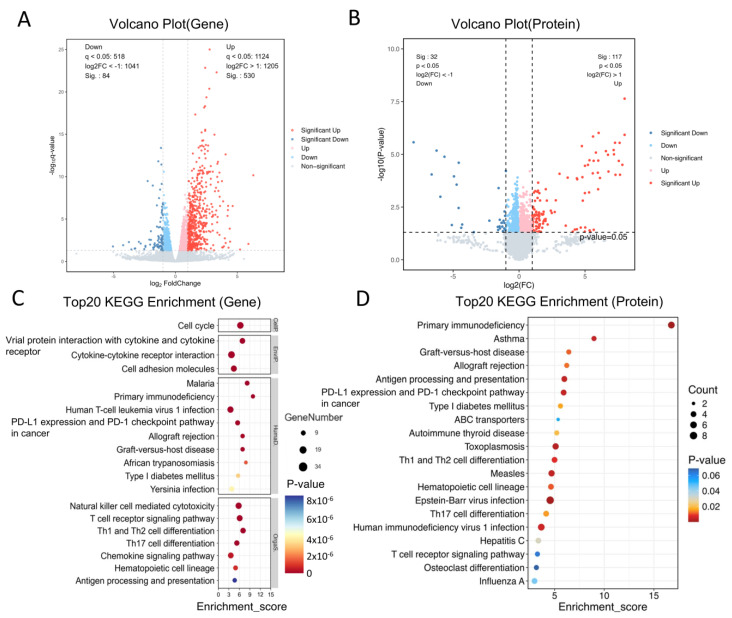
Analysis of differentially expressed genes (DEGs) and proteins (DEPs) between the OVA and Normal groups (*n* = 3). (**A**) The gene volcano plot. (**B**) The protein volcano plot. (**C**) Gene KEGG enrichment analysis. The size of the dot represents the number of enriched genes, and the color represents the significance level. (**D**) Protein KEGG enrichment analysis. The size of the dot represents the number of enriched proteins, and the color represents the significance level.

**Figure 3 molecules-31-00007-f003:**
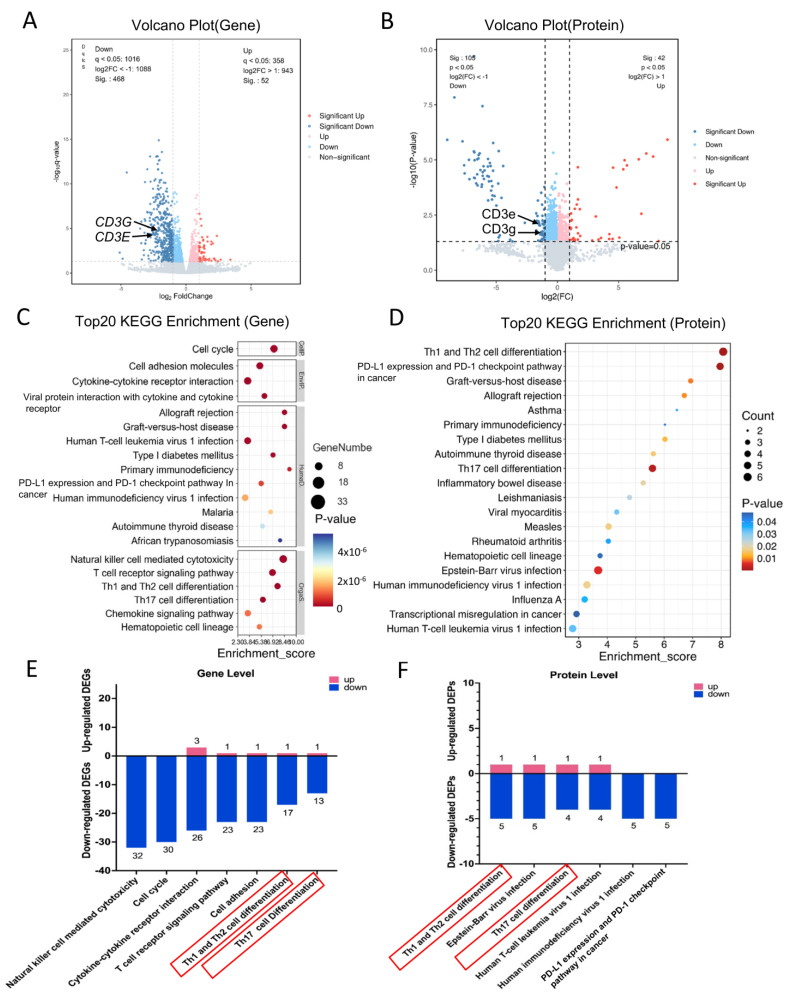
Analysis of DEGs and DEPs between the OVA and ESAT6-Fc groups. (**A**) The gene volcano plot. (**B**) The protein volcano plot. (**C**) Gene KEGG enrichment analysis. The size of the dot represents the number of enriched genes, and the color represents the significance level. (**D**) Protein KEGG enrichment analysis. The size of the dot represents the number of enriched proteins, and the color represents the significance level. (**E**) Expression of DEGs enriched in the key KEGG pathways. (**F**) Expression of DEPs enriched in the key KEGG pathways.

**Figure 4 molecules-31-00007-f004:**
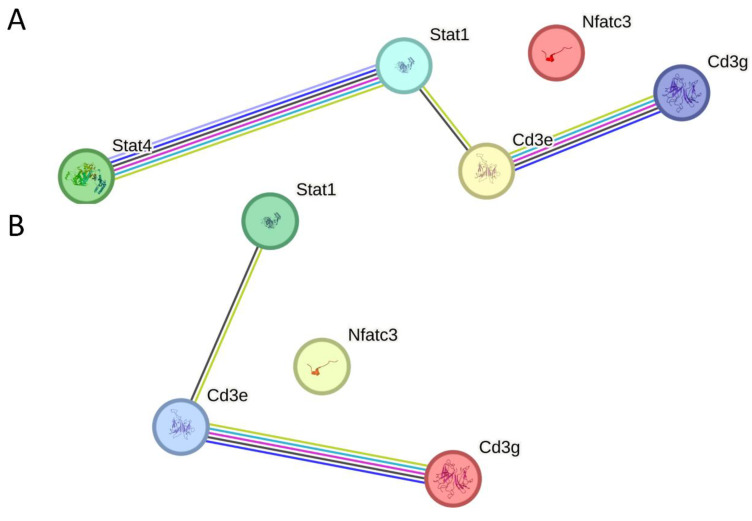
DEGs and DEPs regulated by ESAT6 interaction networks. (**A**) Th1/Th2 cell differentiation. (**B**) Th17 cell differentiation. The KEGG pathways were predicted by https://www.kegg.jp/ (accessed on 19 March 2025).

**Figure 5 molecules-31-00007-f005:**
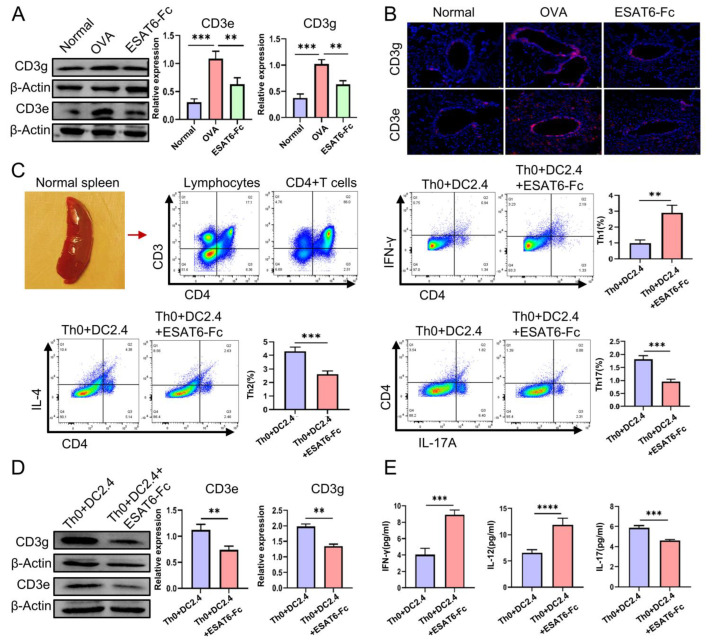
Validation experiments. (**A**) Representative Western Blot images for CD3e and CD3g of mice lung tissue. (**B**) Fluorescence photos for CD3e and CD3g of mice lung tissue. (**C**) Isolation of CD4+T cells from the splenic tissue of healthy mice, and flow cytometry results for the differentiation of Th0 cells into Th1, Th2 and Th17 (**D**) Representative Western Blot images for CD3e and CD3g of CD4+T cells. (**E**) ELISA results for IFN-γ, IL-12, and IL-17A of co-culture system. All data are expressed as the mean ± SD (*n* = 3). ** *p* < 0.01, *** *p* < 0.001 and **** *p* < 0.0001.

## Data Availability

Data will be made available on request.

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
