# Peer review of "Preparation of ESAT6-Fc Fusion Protein and Its Therapeutic Efficacy and Immune Mechanisms in Allergic Asthma Mice via Intranasal Immunization"

_molecules, 2025, doi:10.3390/molecules31010007_

Round 1
Reviewer 1 Report
Comments and Suggestions for Authors
In this study, the authors explore the therapeutic potential of an ESAT6-Fc fusion protein in a mouse model of allergic asthma induced by OVA. Although the concept holds promise, several important concerns need to be addressed before the manuscript can be considered for publication.
<Materials and Methods>
1.The sample size (n = 3–5 mice per group) is too small to support multi-group comparisons with reliable statistical power.
2.The absence of a dose-response analysis for the ESAT6-Fc groups, such as the 5 μg and 10 μg treatments, makes it difficult to determine if ESAT6-Fc played a significant role in the observed immune modulation.
<Results>
3. Fig.1C: The CpG motif is typically employed to stimulate the Th1 response, but there is no evidence supporting its effectiveness in the CpG treatment group.
4. Fig. 1: The authors could assess the levels of IgG2a antibodies and IFN-γ cytokines to verify that ESAT6 plays a role in the differentiation of Th0 cells into Th1 cells in vivo.
5. Fig. 1: The study lacks functional readouts such as airway hyperresponsiveness (AHR) measurement, which is critical for confirming clinical relevance.
6. Fig. 3: The authors might indicate the positions of CD3e and CD3g within the gene or protein volcano plot.
7. Fig. 5: Both DC2.4 cells and T cells used in the coculture system are derived from B6 mice. Therefore, without antigen stimulation, they cannot provoke the activation, proliferation, and differentiation of the T cells.
8. Fig. 5: The findings from in vitro cell culture are consistent; however, they do not demonstrate a dose-response effect with ESAT-6 treatment.
9. Fig. 5: In the cell culture system, the levels of IFN-γ, IL-12, and IL-17 are quite low. I’m curious why the authors didn’t identify the optimal time for collecting the supernatants.
<Discussion>
10. The authors should discuss the enhanced application and therapeutic impact of ESAT6-Fc in comparison to previous ESAT6-like proteins in the context of allergic asthma.
Author Response
Comments 1:The sample size (n = 3–5 mice per group) is too small to support multi-group comparisons with reliable statistical power.
Response1: Thank you for putting this out. This was a typographical mistake in the original text. In fact, there are six mice per group, resulting in a total of 30 mice across five groups. This change can be found in line 109 of the manuscript.
Comments 2:The absence of a dose-response analysis for the ESAT6-Fc groups, such as the 5 μg and 10 μg treatments, makes it difficult to determine if ESAT6-Fc played a significant role in the observed immune modulation.
Response2: Thank you for putting this out,and I agree with this comment. As this was the first study to investigate the effect of ESAT6-Fc on AA, dose selection was primarily informed by previously reported literature. In our future studies, a dose-gradient experimental design will be implemented to further identify the optimal dosage of ESAT6-Fc for immune regulation.
Comments 3:Fig.1C: The CpG motif is typically employed to stimulate the Th1 response, but there is no evidence supporting its effectiveness in the CpG treatment group.
Response3:Thank you for putting this out. In this study, CpG, used as an immune adjuvant, elicited a relatively weak Th1 immune response, showing no statistically significant difference compared to the OVA model group. This finding further suggests that the primary contributor to the Th1 response in the ESAT6-Fc + CpG group is ESAT6-Fc rather than CpG.
Comments 4:Fig. 1: The authors could assess the levels of IgG2a antibodies and IFN-γ cytokines to verify that ESAT6 plays a role in the differentiation of Th0 cells into Th1 cells in vivo.
Response4:Thank you for putting this out. In this study, in addition to measuring the levels of His, LTB4, IgE, IL-4, IL-13, and IL-17A in bronchoalveolar lavage fluid, we also assessed IFN-γ. IFN-γ was not included in the original manuscript due to space constraints in Figure 1; it has now been incorporated. Furthermore, serum IgG and IFN-γ were not evaluated in the current animal study because the immune responses following ESAT6-Fc immunization in normal mice had been previously investigated. Our prior findings demonstrated a significant upregulation of serum IgG, IgG1, IgG2a, and IFN-γ levels, which have already been published in the Chinese Journal of Biologicals.( Yao S,Yang J,Yang Yu,LiH,Wang T, Zhang W,Ma G,Wan Q. Evaluation of the Immunological Efficacy of the Mycobacterium tuberculosis ESAT6-Fc DNA Vaccine. Chin J Biologicals,2023, 36(8): 897–901.).
Comments 5:Fig. 1: The study lacks functional readouts such as airway hyperresponsiveness (AHR) measurement, which is critical for confirming clinical relevance.
Response5: We sincerely appreciate your valuable suggestion. In our subsequent research, we will carefully consider the development of relevant detection methods for airway hyperresponsiveness.
Comments 6:Fig. 3: The authors might indicate the positions of CD3e and CD3g within the gene or protein volcano plot.
Response6: Thank you for putting this out. CD3E and CD3G, along with their corresponding encoded proteins CD3e and CD3g, have been labeled in Figures 3A and 3B, respectively.
Comments 7:Fig. 5: Both DC2.4 cells and T cells used in the coculture system are derived from B6 mice. Therefore, without antigen stimulation, they cannot provoke the activation, proliferation, and differentiation of the T cells.
Response7:In the co-culture system of DC2.4 cells and T cells, ESAT6-Fc was added as described below: “Simultaneously, the ESAT6-Fc fusion protein was introduced into the co-culturing system at a working concentration of 500 ng/mL, and the cells were co-cultured for 24 hours”.Which appears on lines 203 to 204 in the manuscript.
Comments 8:Fig. 5: The findings from in vitro cell culture are consistent; however, they do not demonstrate a dose-response effect with ESAT-6 treatment.
Response8:Thank you for putting this out,and I agree with this comment. In the present study, both animal and cell experiments were conducted using a single dose or working concentration, as determined by reference to existing literature. In future investigations, a dose-gradient experimental design will be adopted to more precisely determine the optimal dosage of ESAT6-Fc for immune regulation.
Comments 9: Fig. 5: In the cell culture system, the levels of IFN-γ, IL-12, and IL-17 are quite low. I’m curious why the authors didn’t identify the optimal time for collecting the supernatants.
Response9: Thank you for putting this out. In cell validation experiments, cell supernatants collected at various time points should ideally be analyzed. However, to maintain consistency with the flow cytometry detection timeline, only the expression levels of IFN-γ, IL-12, and IL-17 following 24-hour stimulation with ESAT6-Fc were assessed.
Comments 10:The authors should discuss the enhanced application and therapeutic impact of ESAT6-Fc in comparison to previous ESAT6-like proteins in the context of allergic asthma.
Response10: Thank you for putting this out,and I agree with this comment. In the discussion section, I elaborated on the finding that both the ESAT6-like protein ESXV and the ESAT6-Fc protein can induce Th1-type cellular immune responses. Although the data suggest that the ESAT6-Fc protein elicits a more pronounced effect on IgE and IFN levels compared to ESXV, a direct comparison cannot be made due to differences in experimental conditions. Therefore, no definitive conclusion was drawn regarding their relative efficacy. In future studies, we plan to directly compare the therapeutic effects of ESAT6-Fc and ESXV-Fc under identical experimental settings.
Reviewer 2 Report
Comments and Suggestions for Authors
The paper is well written and well explained, with testing across different methodologies. The paper is novel, and thanks to the team for your effort and for contributing to this study. The figures are well organized and labeled, and the images are of very good quality for publishing.
Can be published with minor revisions. Please make sure the abbreviations are written only once in the manuscript, and the acronyms are appropriate throughout the publication.
The paper is well written and well explained, with testing across different methodologies. The paper is novel, and thanks to the team for your effort and for contributing to this study. The figures are well organized and labeled, and the images are of very good quality for publishing.
- Atopic Asthma in Introduction Line 62 and Allergic Asthma – both are abbreviated as AA. This isn't very clear in the context. Line 323 AA is abbreviated again. Most of the abbreviations are repeated many times within the manuscript. Please ensure that there is no repetition of the abbreviation, as this has increased the word count, and repetitive naming does confuse and makes it difficult to be readable to the researchers.
- Line 234- What is Th17A? This is not clarified in the context. Is it confused with IL-17A?
- Why was intranasal administration preferred over the intravenous or other pathways? Can you explain a few sentences about this in the introduction, abstract, and the final discussion part?
- Did you perform the treatment and assays with extended incubation times to study the effect of the ESAT6 fc fusion proteins? You have described in the discussion that you have to determine the safety of this drug by extended treatments.
- Can you explain better about why CpG adjuvants were studied in the mice study in the section 2.2, Line 106.. There are no explanations in the Introduction regarding this and it looks like it was jumped to the point straight without explanation.
- In the results section, section 3.1, can the paragraphs be separated while explaining Figure 1.c. and 1.d. for more clarity as it has a lot of information.
- Some western blotting original images in the file is not of good quality, it can be improved.
Author Response
Comments 1: Atopic Asthma in Introduction Line 62 and Allergic Asthma – both are abbreviated as AA. This isn't very clear in the context. Line 323 AA is abbreviated again. Most of the abbreviations are repeated many times within the manuscript. Please ensure that there is no repetition of the abbreviation, as this has increased the word count, and repetitive naming does confuse and makes it difficult to be readable to the researchers.
Response 1: Thank you for putting this out. Allergic asthma (AA) has undergone a comprehensive revision.
Comments 2: Line 234(now 243)- What is Th17A? This is not clarified in the context. Is it confused with IL-17A?
Response 2: Thank you for putting this out.Yes, here it is confused with IL-17A,and has been corrected to Th17 in Line 248.
Comments 3: Why was intranasal administration preferred over the intravenous or other pathways? Can you explain a few sentences about this in the introduction, abstract, or the final discussion part?
Response 3: Thank you for putting this out. The advantages of intranasal immunization with ESAT6-Fc have been added in the introduction, specifically in lines 85–86.
Comments 4: Did you perform the treatment and assays with extended incubation times to study the effect of the ESAT6 fc fusion proteins? You have described in the discussion that you have to determine the safety of this drug by extended treatments.
Response 4: Thank you for putting this out. Currently, we are conducting the second phase of animal experiments to further evaluate the long-term effects of ESAT6-Fc intranasal immunization on OVA-induced AA mice ,and to assess its safety profile.
Comments 5: Can you explain better about why CpG adjuvants were studied in the mice study in the section 2.2, Line 106(now109).. There are no explanations in the Introduction regarding this and it looks like it was jumped to the point straight without explanation.
Response 5: Thank you for putting this out.The CpG group was explicitly designated as a CPG-ODN immune adjuvant.
Comments 6: In the results section, section 3.1, can the paragraphs be separated while explaining Figure 1.c. and 1.d. for more clarity as it has a lot of information.
Response 6: Thank you for putting this out. According to your suggestion, the elaboration of Figure 1.C and Figure 1.D has been separated in the manuscript .
Comments 7: Some western blotting original images in the file is not of good quality, it can be improved.
Response 7:Thank you for putting this out,and I agree with this comment. The student conducting the WB is a novice with limited experience. However, she is highly diligent and demonstrates strong commitment to her work. I am confident that her WB skills will continue to improve over time.
Round 2
Reviewer 1 Report
Comments and Suggestions for Authors
I have no comments regarding this manuscript and accept it in its current form.